# School Disengagement Predicts Accelerated Aging among Black American Youth: Mediation by Psychological Maladjustment and Moderation by Supportive Parenting

**DOI:** 10.3390/ijerph191912034

**Published:** 2022-09-23

**Authors:** Mei Ling Ong, Eric T. Klopack, Sierra Carter, Ronald L. Simons, Steven R. H. Beach

**Affiliations:** 1Center for Family Research, The University of Georgia, Athens, GA 30605, USA; 2Leonard Davis School of Gerontology, University of Southern California, Los Angeles, CA 90089, USA; 3Department of Psychology, Georgia State University, Atlanta, GA 30303, USA; 4Department of Sociology, The University of Georgia, Athens, GA 30605, USA; 5Department of Psychology, The University of Georgia, Athens, GA 30605, USA

**Keywords:** school disengagement, psychological maladjustment, supportive parenting, Black Americans, aging

## Abstract

Early experiences of school disengagement may serve as a warning sign for later young adult adjustment difficulties and eventually contribute to accelerated aging among Black American youth. At the same time, supportive parenting may play a protective role. Using longitudinal data from the Family and Community Health Study (FACHS), we examined psychological maladjustment (comprising depression, lack of self-regulation, and low self-esteem) as a mediator of the relationship between school disengagement and accelerated aging. We also examined the effect of supportive parenting in buffering the impact of school disengagement on adulthood outcomes by controlling for covariates. Hypotheses were examined in a sample of 386 (Mean age = 28.68; Females = 62.7%; Males = 37.3%) Black American youth who were followed into young adulthood. Path modeling was used to test hypothesized relationships. We found school disengagement, i.e., problems with school attendance, performance, and engagement, reported across ages 10–18, predicted psychological maladjustment, which, in turn, predicted accelerated aging at age 29. We also found a buffering effect for supportive parenting. No significant gender difference in the indirect effect or buffering effect was found. This study highlights the potential importance of greater attention to school disengagement to identify and potentially influence long-term health trajectories and adult outcomes for Black American youth.

## 1. Introduction

Education and health have been found to be reliably related, with higher academic achievement associated with better health outcomes [1]. Likewise, poor academic achievement and school dropout forecast increased the likelihood of a number of adverse, stressful adult outcomes [2], such as poverty [3], later unemployment [4], and arrest [5]. In addition, youth with poor academic achievement and school dropout have an increased risk for a range of adverse physical health outcomes (e.g., cardiometabolic risk), leading some to call for greater attention to educational barriers as a way to reduce health disparities [6].

An academic achievement gap exists across ethnic minority groups, especially among Black Americans. Black American children often experience challenging circumstances that can affect their academic performance, including poverty, community violence, and parents with low education levels. These can affect absenteeism in early schooling, resulting in poor academic achievement and a seriously increased risk of school dropout [1,7,8,9]. Black Americans are also at increased risk of school difficulties due to cultural conflicts and language differences. In addition, Black Americans have been historically unfairly denied opportunities for education; therefore, they protect themselves from the dominant group’s self-devaluation by rejecting the dominant group’s values and devaluing school [10]. This defensive action can sometimes turn into attitudes and behaviors that Ogbu [10] called a low-effort syndrome, i.e., lack of perseverance in academic striving and negative academic attitudes in school. Thus, Black Americans are likely to face multiple adversities [11] that can compromise their school experience, perhaps setting the stage for greater risk of various health-related problems in adulthood [12].

School dropout in the Black American adolescent period is elevated, posing a severe and critical issue for various socioeconomic outcomes, such as job instability [13] and economic difficulties [14]. It is crucial to probe the impact of school problems and school disengagement as it is one of the factors influencing school dropout [15]. School disengagement is characterized by adolescents who do not feel belonging to the school, lack motivation in learning, have academic disidentification, or always skipping or not attending school [16,17]. Likewise, school problems and school disengagement are potential predictors of mental health problems in young adulthood, such as depression and anxiety [4], and substance use [18]. Unfortunately, the literature on the utility of school experiences as a predictor of later physical and mental health problems is sparse. It is essential for Black American youth due to the disparate rates of school dropout and experiences of discrimination compared to other racial groups [19]. Accordingly, school difficulties, such as school disengagement, may serve as a warning sign, forecasting a developmental trajectory requiring attention and intervention.

School disengagement can also raise the risk for internalizing problems, such as depression [20] and other related issues in young adulthood, including reduced self-regulation [21] and lower self-esteem [8]. Accordingly, developing a constellation of depression, self-regulation difficulties, and self-esteem problems in young adulthood may be one mechanism linking school disengagement with later health problems [22,23,24]. However, supportive parenting (warmth, acceptance, and involvement) may be a resilience factor, potentially buffering youth against internalizing and externalizing risks associated with disengagement from school. Parental involvement is associated with reduced behavior problems [19,25,26]. Supportive parenting may also buffer the effects of school disengagement on psychological maladjustment by moderating the effect of school disengagement, creating an indirect buffering effect on the pathway from school disengagement to psychological maladjustment and ultimately health problems. 

Path modeling and a sample of 386 Black Americans are used to investigate the impact of school disengagement on accelerated aging among Black Americans and examine whether it is mediated through psychological maladjustment and buffering by supporting parenting.

### 1.1. Accelerated Aging Measure

To optimally examine the impact of childhood difficulties on young adult health, it is crucial to overcome problems associated with self-reported health and the long delay associated with waiting for chronic illness to emerge. In the current study, we utilize an mRNA measure of transcriptomic aging developed by Peters et al. [27], comprising 1497 sites identified as being related to chronological age. Peters et al. [27] reported strong correspondence between the discovery sample of European ancestry and the replication sample comprising various ethnic groups, including African Americans, Native Americans, and Hispanics. Corresponding to earlier epigenetic clock measures of Horvath [28] and Hannum et al. [29], the transcriptome index is highly correlated with age and the difference between an individual’s predicted and chronological age indicates in years, the extent to which they are experiencing accelerated or decelerated aging. By examining the deviation of chronological age from predicted age based on the 1497 mRNA sites, it is possible to determine the extent to which a given individual is “accelerated” in their aging. The advantages of utilizing an index based on the transcriptome are multifaceted. First, the sites used in the transcriptome index are in gene pathways with links to aging, including gene pathways related to metabolic functioning, immune senescence, and mitochondrial decline. In addition, the transcriptome index directly measures gene expression with functional significance for aging [30] and is proximal to biological outcomes of interest.

Moreover, it is related to metabolic functioning, immune senescence, DNA repair, inflammation, and mitochondrial effectiveness, implicating the development of cardiometabolic problems and chronic illnesses, such as diabetes, coronary heart disease, stroke, and Alzheimer’s [27]. This evidence indicates that accelerated aging, assessed by the transcriptome index, consists of many gene expression changes that are associated with the onset of chronic illness [31]. Consequently, this measure of accelerated aging is appropriate to capture potential processes of interest regarding the risk for future health problems. It is not confounded by shared response biases with either school disengagement or young adult adjustment, suggesting observed relationships will provide a conservative estimate of their association with health.

### 1.2. Psychological Maladjustment as a Mediator

Psychological factors, such as mental health problems, are related to persistent school disengagement [4,32]. Premature school disengagement can increase the risk for depression [20], and adolescents who are disengaged from academics at an early age have fewer friends, lower academic self-concepts, lower levels of self-esteem [32], and poorer self-regulation than peers [33]. Accordingly, school disengagement may lead to a constellation of psychological factors, including depression, difficulties with self-regulation, and low self-esteem, that may forecast later health problems.

Strong evidence [4,34] suggests depression is a critical factor during childhood and adolescence because it can influence a number of health outcomes. Depression beginning in the teenage years can affect the quality of life [35] and increase the risk for lifetime adverse psychological outcomes, including substance use, suicide, anxiety, and school problems [19]. Depression may also lead to chronic negative beliefs about ability [36], reinforce low self-efficacy, and reinforce lack of self-regulation, leading to a constellation of negative psychological symptoms and decreasing school engagement.

Conversely, youths with good self-regulation are likely to be healthier [23,37]. In contrast, those with poor self-regulation have a higher risk of health and behavior problems [23], including drug use [34], theft, risky sex, and gambling [23,38]. In addition, adolescents who are lower in self-regulation have greater problems with cardiovascular disease risk [39] and high cholesterol [38], as well as increased early mortality [40], indicating depression and low self-regulation in early life are associated with school disengagement and adverse health outcomes in adulthood.

Low self-esteem—thoughts and feelings about an individual’s self-worth or ability [24,41]—is also an important psychological factor that affects school disengagement and healthy aging [22]. Teenagers with low self-esteem are more likely to have difficulties identifying helpful coping strategies [42], amplifying the effect of school problems on later health outcomes [24,43,44]. Furthermore, because low self-esteem is strongly associated with anxiety and depression [45], it is reasonable to include it in an overall index of negative psychological symptoms, potentially mediating the association between school disengagement and young adult health. Accordingly, we constructed psychological maladjustment using three indicators: depression, lack of self-regulation, and low self-esteem.

### 1.3. Supportive Parenting as a Moderator

Parent–child relationships and parents’ involvement in their children’s activities influences have emerged as a crucial element in models identifying potential intervention points for early prevention of school disengagement and increased resilience to psychosocial stress (e.g., [37,41,46]). Parenting styles, including authoritarian, authoritative, and permissive domains [47], have been shown to relate to teenagers’ psychological problems, behavioral problems, and academic performance (e.g., [47,48]). Authoritative parenting emphasizes warmth, involvement, responsiveness, and encouraging autonomy. Permissive parenting makes few demands, exhibits noncontrolling behaviors, and uses minimal punishment. Authoritarian parenting characterizes a high level of directive, restriction and rejection, and power-asserting behaviors [47]. Supportive parenting (i.e., authoritative parenting) seems to be superior at fostering higher academic performance [26,46,48], healthier psychological adjustment, and lower rates of behavioral problems [25]. Teenagers who perceive academic parental involvement at home receive higher grades and have increased academic motivation [26,46]. Likewise, supportive parenting has been linked to improved psychological factors in adolescence [7,37,46]. For instance, Frazer and Fite [19] found that parents’ involvement in their children’s lives was negatively related to depression symptoms and may buffer the impact of adverse experiences on depression. Supportive parenting is positively related to self-esteem [49] and cognitive and emotional self-control [7]. For example, when parents and teenagers engage in supporting interactions, teenagers learn their parents’ cognitive control and problem-solving skills through observation and modeling [7,50]. Beach and colleagues [51,52] found supportive parenting can buffer the psychological stress response of children and teenagers, as well as decreasing accelerated aging [53]. Together, these findings [7,51,52,53] suggest supportive parenting during early adolescence may provide an important buffer of effects from disengagement from school to accelerated aging in Black American youth by reducing the impact of psychological maladjustment. To address these issues, we examine three hypotheses in the next section.

### 1.4. Hypotheses

Based on the preceding literature we formulated three specific hypotheses reflecting our expectation of longitudinal associations between school disengagement and problems manifesting in young adulthood and the potential protective role of parenting: 

**Hypothesis** **1.**
*Youth psychological maladjustment will mediate the relationship of school problems and disengagement in adolescence with accelerated aging in young adulthood even after controlling confounding variables. This would be reflected by a significant indirect effect from school disengagement to accelerated aging through young psychological maladjustment.*


**Hypothesis** **2.**
*The association of school disengagement with psychological maladjustment will be buffered by supportive parenting. Higher levels of supportive parenting will diminish the relationship between school disengagement and psychological maladjustment.*


**Hypothesis** **3.**
*If hypotheses 1 and 2 are supported, this will lead to a significant buffering effect of the indirect pathway from school disengagement to accelerated aging by supportive parenting.*


## 2. Materials and Methods

### 2.1. Sample

The hypotheses were examined using data from seven waves of the longitudinal Family and Community Health Study (FACHS IV; reviewed by the University Institutional Review Board of the University of Georgia), a multi-site investigation of neighborhood and family effects on mental health and development of Black American children in Iowa and Georgia [53]. The FACHS sample consisted of 889 Black American families at wave 1 in 1997–1998. All families were in 5th grade at the time of recruitment. The sampling strategy was designed to generate a data set with families representing a range of socioeconomic statuses and a wide variety of neighborhood settings. Families were recruited from neighborhoods that varied on demographic characteristics, precisely racial composition (percentage African American) and economic level (percentage of families with children living below the poverty line). In Iowa and Georgia, households were randomly selected from the sampling frame using rosters of fifth-grade students in the public-school systems. At the first wave, 422 families resided in Georgia and 467 resided in Iowa. All children were in the fifth grade and averaged 10 (SD = 0.63) years of age. The majority of primary caregivers were females. Approximately 19% had less than a 12th grade education. Almost 65% lived in large urban areas, 15% lived in the suburbs, and 21% lived in rural areas. There were 779 interviews at wave 2 in 1999–2000, 767 at wave 3 in 2001–2002, 714 at wave 4 in 2004–2005, 689 at wave 5 in 2007–2008, and 669 at wave 6 in 2010–2011. Data collection was completed that included blood draws at wave 7 in 2015–2016. Given the logistics of scheduling home visits by phlebotomists, only members of the sample residing in Georgia, Iowa, or a contiguous state were identified as eligible. Thus, after excluding participants who were deceased, incarcerated, or unreachable for some reason, we were left with a pool of 545 individuals, and 470 (86%) participants agreed to be interviewed and provided blood. After excluding samples with poor quality (*n* = 81) and no amplification (*n* = 3), successful assays for mRNA expression were obtained for 386 participants (Mean age = 28.68; females = 242 and males = 144) who were not missing data on other variables of interest. The average annual income of young adults was $22992.75 (SD = 17346.44). 

### 2.2. Measures

*School Disengagement*. Participants had a mean age of 10.41, 12.29, 15.67, and 18.77 at Waves 1–4, they responded to a seven-item scale developed for FACHS [54] that asks respondents to indicate how much they agree (strongly disagree = 1, strongly agree = 4) with various items such as “School bores you,” “I have been in trouble for skipping or not attending school,” and “You do not do well at school”. The four items that reflected positive experiences with school were reverse coded so that higher scores always indicated greater school disengagement. Score was averaged across items to form a measure of school disengagement. Cronbach’s alpha was 0.61 at Wave 1, 0.66 at Wave 2, 0.69 at Wave 3, and 0.65 at Wave 4.

*Psychological maladjustment*. Three scales (depression, poor self-regulation, and low self-esteem) were each averaged across waves 5–7 when the participants had a mean age of 21.53, 23.53, and 28.68. The mean of the three standardized scale scores formed the index of psychological maladjustment across young adulthood. *Depression* was assessed using nine items from the University of Michigan Composite International Diagnostic Interview UM-CIDI; [55] measure of depressive symptoms. Respondents reported symptoms of depression (0 = no; 1 = yes) with items such as “felt sad, empty, or depressed most of the day” for at least a 2-week period in the past year. Cronbach’s alpha was 0.78 at Wave 5, 0.83 at Wave 6, and 0.86 at Wave 7. *Lack self-regulation* was assessed using 11 items from Kendall and Wilcox’s [56] self-regulation scale. Cronbach’s alpha was 0.71 at Wave 5, 0.73 at Wave 6, and 0.84 at Wave 7. *Low self-esteem* was evaluated using five items from a revised version of the Rosenberg’ self-esteem Scale [57,58]. Respondents reported agreement with items such as “At times I think I am no good at all.” Cronbach’s alpha was 0.74 at Wave 5, 0.77 at Wave 6, and 0.76 at Wave 7.

*Transcriptional Index of Biological Age*. *n* = 386 usable Blood samples were sent to the Rutgers repository. Samples were annotated using HumanHT-12 from Illumina, the 47,323 probes were filtered to remove probes with a detection threshold of *p* ≤ 0.05, leaving 44,846 probes for analysis. Following quantile normalization, the data were log2 transformed. Accelerated aging was measured using the mRNA measure developed by Peters et al. [27]. Publicly available software (https://trap.erasmusmc.nl, accessed on 22 June 2022) was used to compute Transcriptomic age, and the mean biological age was 29.17 (SD = 0.76). We regressed biological age on chronological age and used the residual score as our measure of accelerated aging [31]. These residuals had a mean of zero to represent both positive and negative deviations from mean age acceleration (in years). Positive scores indicate accelerated aging. 

*Supportive Parenting*. Supportive parenting was assessed using a nine-item scale that was adapted from instruments developed for the Iowa Youth and Families Project IYFP; [59]. At Waves 1–4, respondents were asked to report whether descriptors are 4 = always, 3 = often, 2 = sometimes, or 1 = never occurred with items such as “acted supportive and understanding toward you,” and “told them that she or he loves them.” Score was averaged across items to form a measure of supportive parenting. Higher scores always indicated higher levels of supportive parenting. Cronbach’s alpha was 0.83 at Wave 1, 0.89 at Wave 2, 0.90 at Wave 3, and 0.92 at Wave 4.

*Covariates*. Several studies [54,60,61] have found that a variety of health risk behaviors, such as diet, as well as demographic variables such as education and socioeconomic status (SES) risk, affected health outcomes. Thus, in this study we controlled gender, education, SES risk (at Wave 1), and health-behavior covariates at Wave 7 (age 29) to reduce the probability of spurious effects. *Gender* (male = 1) was controlled in all analyses and was examined in exploratory analyses as a potential moderator of response. *Healthy diet* was assessed using two items about the frequency of fruit and vegetable consumption during the previous 7 days, and *physical exercise* items evaluate the amount and intensity of activity over the past 7 days. *Alcohol consumption* was measured by assessing the frequency of binge drinking over the past 12 months. *Smoking behavior* was measured by evaluating the frequency of cigarettes smoked in the last 3 months. *Education* was reported as the highest level of education completed. *Socioeconomic status* (SES) risk was measured by averaging caregivers’ reports of five indicators: (a) noncompletion of high school or equivalent; (b) current unemployment, (c) single-parent family structure, (d) current family receipt of Temporary Assistance for Needy Families, and (e) income rated by the primary caregiver as currently inadequate to meet all needs. Each indicator was scored dichotomous (0 = absent, 1 = present), with the sum indicating level of risk.

### 2.3. Analytic Strategy 

We performed path modeling in M*plus* 8 [62] to examine the indirect effect of early school disengagement on accelerated aging through young adult psychological maladjustment and to examine the stress-buffering effects by supportive parenting. We first examined the direct effect of school disengagement across Wave 1 through 4 on accelerated aging, averaging scores at Waves 1 to 4 to form the school disengagement measure. Next, using path modeling to assess the indirect effect, *Mplus* was used to generate a bootstrapped confidence interval (CI). The significance of hypothesized indirect effects was assessed by examining the 95% confidence interval (CI) estimated with bias-corrected and accelerated bootstrapping with 1000 replications. To evaluate the goodness-of-fit of the model, we report the standardized root mean square residual (SRMR < 0.05) and the comparative fit index (CFI > 0.90) as well as the chi-square test and degrees of freedom. Third, we tested the moderated mediation hypotheses using path modeling. We examined the interaction of school disengagement and supportive parenting in predicting psychological maladjustment. After averaging items across items to form measures of school disengagement and supportive parenting, the measures were standardized (i.e., a mean of 0 and a standard deviation of 1). The school disengagement and supportive parenting measures were standardized to allow calculation of interaction terms that would not be correlated with the main effects, reducing potential for multicollinearity among variables in the model [60]. In addition, standardized scores make it easier to read the results [63], easier to interpret the simple slopes, and easier to test slopes [64]. To explicate the shape of a significant interaction effect, we plotted the simple slopes with school disengagement and psychological maladjustment, showing the effect of school disengagement on psychological maladjustment with confidence intervals around the regression lines for high vs. low supportive parenting and reporting conditional indirect effects. Finally, we investigated gender differences using the multiple group analysis option in *Mplus*. We included gender, education, healthy diet, exercise, alcohol consumption, and smoking as covariate variables in all analyses. We compared those retained for our analyses (*n* = 386) with those who were lost to attrition (*n* = 503) on all study variables used at baseline. We found that only gender was significantly different for those lost to attrition, with more males than females lost to attrition.

## 3. Results

### 3.1. Descriptive and Association Analysis

The mean, standard deviation, and zero-order correlations for the study variables are shown in Table 1. As expected, school disengagement in childhood was positively correlated with young adult psychological maladjustment (*r* = 0.244, *p* = < 0.001) and accelerated aging (*r* = 0.101, *p* = 0.048), negatively correlated with supportive parenting (*r* = −0.337, *p* =< 0.001) and education (*r* = −0.217, *p* =< 0.001). In addition, accelerated aging was positively related to psychological maladjustment (*r* = 0.185, *p* =<0.001), negatively related to supportive parenting (*r* = −0.133, *p* = 0.009), and smoking (*r* = −0.135, *p* = 0.008). Moreover, psychological maladjustment was negatively related to supportive parenting (*r* = −0.253, *p* =< 0.001), positively related to alcohol consumption (*r* = 0.194, *p* =< 0.001), and smoking (*r* = 0.108, *p* = 0.033). School disengagement and accelerated aging were positively associated with depression, poor self-regulation, and low self-esteem, and supportive parenting was negatively correlated to them, as shown in Table 2.

### 3.2. Test of Mediation Model

We first tested a baseline model, evaluating the direct effect of school disengagement on accelerated aging at age 29. Using bootstrapping in M*plus* with 1,000 replications, school disengagement was a significant predictor of accelerated aging at age 29 (*β* = 0.104, *t* = 1.99, CI = [0.015, 0.199]), even after controlling for gender (male = 1), education, healthy diet, exercise, alcohol consumption, smoking, and socioeconomic status (SES) risk.

Addressing hypothesis 1, the association of school disengagement and problems in childhood with psychological maladjustment in young adulthood was significant and positive (*β* = 0.236. *t* = 4.72, CI = [0.118, 0.338]), and, in turn, psychological maladjustment was associated with accelerated aging (*β* = 0.190, *t* = 3.57, CI = [0.101, 0.281]), even after controlling for covariates. Prior to including the hypothesized mediator, there was a significant direct effect of (*β* = 0.104, 95% CI = [0.015, 0.199]). When psychological maladjustment was added as a mediator between school disengagement and accelerated aging, however, the direct association between school disengagement and accelerated aging was reduced from 0.104 to non-significance (*β* = 0.059, *t* = 1.12, 95% CI = [−0.038, 0.154]). These associations are shown in Figure 1.

As expected, the significant associations from school disengagement/problems to psychological maladjustment and then to accelerated aging resulted in a significant indirect effect of school disengagement on accelerated aging through psychological maladjustment. The 95% CI around the estimated indirect effect did not contain zero, IE = 0.045, 95% CI [0.020, 0.084]. The total effect was also significant (IE = 0.104, 95% CI [0.015, 0.199]).

There was no evidence of multicollinearity among the study variables because diagnostic variance inflation factor scores for all variables in the regression were below 10, ranging from 1.07 to 1.17, and tolerance was greater than 0.1, ranging from 0.85 to 0.93. 

### 3.3. Test of Moderated Mediation Model

To address the hypothesized stress-buffering effects of supportive parenting, we created an interaction term by multiplying school disengagement by supportive parenting and examining it as a predictor of psychological maladjustment after controlling gender (male =1), education, healthy diet, exercise, alcohol consumption, smoking, and SES risk. As predicted, the interaction effect of school disengagement and supportive parenting was negatively associated with psychological maladjustment, *β* = −0.113, *t* = −2.29, CI = [−0.227, −0.005], consistent with a significant buffering effect. To explicate the significant interaction, we graphed the simple slopes for the relationship between school disengagement and psychological maladjustment at high (i.e., 1 standard deviation (SD) above the sample mean) and low (i.e., 1 SD below the sample mean) supportive parenting as is shown in Figure 2. 

Consistent with the stress-buffering hypothesis, Figure 2 demonstrates that the effect of school disengagement on young adult psychological maladjustment with high supportive parenting was not significant (*b* = 0.134, *p* = 0.444). In contrast, the effect of school disengagement on psychological maladjustment for youth with low levels of supportive parenting was strong and positive (*b* = 0.579, *p* ≤ 0.001), supporting Hypotheses 2.

Given that supportive parenting buffered the effect of school disengagement on psychological maladjustment, we also employed a bootstrapping technique in *M**plus* with 1000 replications to evaluate the significance of the conditional indirect effect. The moderated impact of school disengagement on psychological maladjustment and significant indirect effects on accelerated aging are shown in Figure 3. The moderated mediation model provided a good fit to the data, χ^2^= 5.881, *df* = 2, *p* = 0.0528; CFI = 0.952 and SRMR = 0.023. As expected, school disengagement had a significant indirect effect on accelerated aging at age 29 through psychological maladjustment among those with low supportive parenting (IE = 0.134, 95% CI [0.054, 0.260]). Still, there was no significant effect among individuals with high supportive parenting (IE = 0.031, 95% CI [−0.060, 0.114]). Hence, the pattern of study supports hypothesis 3 that the indirect effect of school disengagement is conditional on levels of supportive parenting.

Finally, to examine for any possible sex effects, multiple group analyses (male vs. female) were conducted. We compared models that constrained all paths to be equivalent for males and females with models that allowed one path at a time to differ freely. We examined chi-square change with one degree of freedom. The results showed no paths were significantly different as a function of sex (school disengagement → psychological maladjustment: Δ$\chi_{\left(1\right)}^{2}$ = 0.264, *p* = 0.607; psychological maladjustment → accelerated aging: Δ$\chi_{\left(1\right)}^{2}$ = 1.441, *p* = 0.230), indicating no significant sex differences in the effects of school disengagement on accelerated aging through psychological maladjustment. 

## 4. Discussion

Prior research suggests early adversity can contribute to health consequences much later in life, contributing to racial disparities in morbidity and mortality [65]. One crucial but understudied source of childhood stress and adversity may be children’s experiences at school. School disengagement and problems may be an important early predictor of mental and physical well-being over time, making it especially important to track and attend to them. Given the well-documented link between education level and mental health (e.g., [2,4,15]), we hypothesized school disengagement might presage long-term health consequences, and supportive parenting would be helpful in countering the adverse effects of school disengagement. Numerous studies (e.g., [7,46,49,53]) have found that supportive parenting is protective against psychological maladjustment. For example, adolescents receiving greater supportive parenting perform better academically and have higher self-esteem [19,49]. However, to our knowledge, no study has explored the relationship among disengagement from school, supportive parenting, psychological maladjustment, and accelerated aging. The present investigation tests a stress-buffering model using multiple sources of data, including self-report measures of school disengagement, supportive parenting, psychological maladjustment, and an mRNA measure of accelerated aging. 

Our results indicated school disengagement may exert effects on health that go beyond their association with educational attainment, health behaviors, and income. School disengagement in childhood (age 10–18) was associated with accelerated aging in adulthood (age 29), even after controlling for gender, education level, healthy diet, exercise, alcohol consumption, smoking, and socioeconomic status. Our results suggested that psychological maladjustment was a mediator of the effect of school disengagement and accelerated aging in young adulthood, implying that psychological maladjustment is a precursor of long-term health problems. Moreover, consistent with the stress-buffering model [66,67], we attempted to determine whether supportive parenting could buffer the association between school disengagement and young adult psychological maladjustment. Our results supported the hypothesis that high levels of supportive parenting received in adolescence would moderate the effect of school disengagement on psychological maladjustment, extending prior intervention research showing supportive parenting can buffer the impact of early adversity and maltreatment on health outcomes [67]. Indeed, school disengagement only had a significant positive relation to young adult psychological maladjustment in the context of low levels of supportive parenting during adolescence. These findings were evident across each measure of school disengagement considered separately. Our findings further indicated no sex differences in the relationships among school disengagement, psychological maladjustment, and accelerated aging. 

The current research has some limitations. First, the sample was not nationally representative because all participants in the present sample lived in towns and small cities in Georgia and Iowa. Replication using other, more broadly representative samples of Black Americans to predict accelerated aging from school disengagement would add confidence to these results. Second, this study assessed DNA gene expression at a one-time point; future studies, including assessment of accelerated aging at more time points, would allow better examination of effects on stability and change across young adulthood. Third, our finding on school disengagement effects on accelerated aging should be replicated for other marginalized groups who experience disparate rates of school dropout and/or school disengagement. The current sample comprised Black American families, so future studies should include other ethnic groups to examine potential racial, ethnic, or other differences more fully in the association of school disengagement with later health outcomes. Finally, the data cannot distinguish between causal effects and predictive effects. Thus, future work to examine the impact of interventions designed to decrease early school disengagement would be helpful in clarifying causal relationships in the model. Likewise, examination of interventions that increase supportive parenting would be useful to see if they are successful in reducing the association between school problems and later adjustment and accelerated aging.

## 5. Conclusions

The current results provide evidence that early experiences of school disengagement may serve as a warning sign for later young adult adjustment difficulties and eventually accelerated aging among African American youth. This problematic trajectory may, in turn, set the stage for health problems later in life. Therefore, our findings provide evidence supporting increased surveillance of these difficulties and increased efforts to utilize this information to better address school disengagement and better prevent future psychological difficulties and accelerated aging among Black American youth. Future research should consider additional biopsychosocial mechanisms and health behaviors that may also play a role in connecting school disengagement with later mental and physical health problems. Likewise, future research should consider additional potential moderators, including community and peer influences. Implications for School Health Policy, Practice, and Equity. 

Despite its limitations and the need for future studies, the current study provides a better understanding of the link between school disengagement and accelerated aging in adulthood for Black Americans. Our findings imply that school disengagement is a critical adverse experience associated with later accelerated aging and psychological maladjustment. Our findings also suggest that supportive parenting plays a vital role in buffering against the potential threat of painful experiences leading to psychological dysfunction.

This finding is consistent with prior research indicating Black American parents play an important role in supporting their children in school contexts that devalue their academic progress [68,69], and so provides support for the development of programs to reduce school disengagement as well as support for the use of parenting interventions in the school setting to influence long-term outcomes resulting from school disengagement. These may have the potential to result in better health outcomes in young adulthood and beyond for Black American youth.

## Figures and Tables

**Figure 1 ijerph-19-12034-f001:**
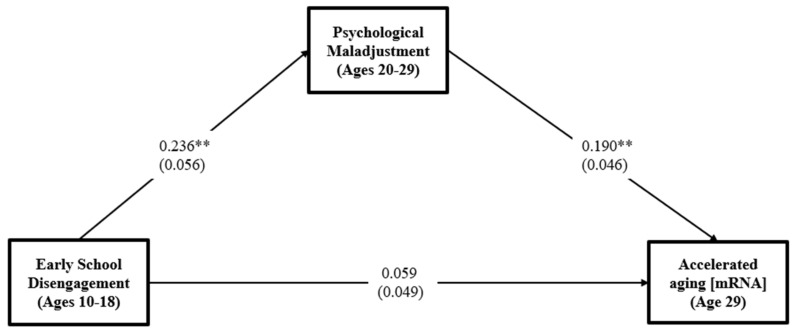
Effects of school disengagement on accelerated aging at age 29 through psychological maladjustment. The model is saturated. Values are standardized parameter estimates, and standard errors are in parentheses. Gender, education, healthy diet, exercise, alcohol consumption, and smoking are controlled in these analyses. ** *p* ≤ 0.01, * *p* ≤ 0.05 (two-tailed tests), *n* = 386.

**Figure 2 ijerph-19-12034-f002:**
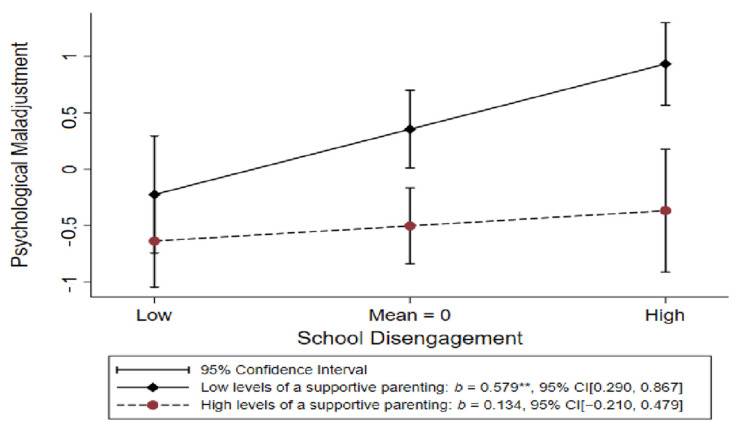
The simple slope test of school disengagement on psychological maladjustment by level of supportive parenting. The lines represent the low (i.e., one SD below the mean) and high (i.e., one SD above the mean) levels of supporting parenting. ** *p* ≤ 0.01, * *p* ≤ 0.05, (two-tailed tests).

**Figure 3 ijerph-19-12034-f003:**
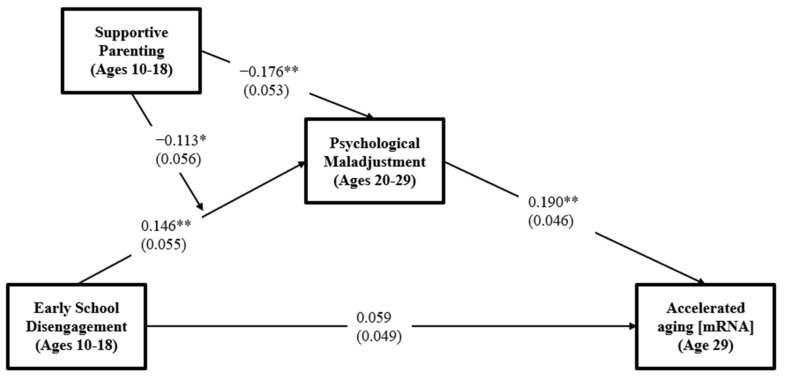
The conditional indirect effects of school disengagement and its interaction on accelerated aging at age 29 through psychological maladjustment moderated by supportive parenting. Chi-square = 5.881, *df* = 2, *p* = 0.0528; CFI = 0.952; SRMR = 0.023. Values are standardized parameter estimates, and standard errors are in parentheses. Gender, education, healthy diet, exercise, alcohol consumption, and smoking are controlled in these analyses. ** *p* ≤ 0.01, * *p* ≤ 0.05 (two-tailed tests), *n* = 386.

**Table 1 ijerph-19-12034-t001:** Correlation, Mean, and Standard Deviations for the Study Variables (*n* = 386).

Variables.	1	2	3	4	5	6	7	8	9	10	11
1. Accelerated Aging	—										
2. School Disengagement	0.101 *	—									
3. Psy Maladjustment	0.185 **	0.244 **	—								
4. Supportive Parenting	−0.133 **	−0.337 **	−0.253 **	—							
5. SEX (Male =1)	−0.090 ^†^	0.077	−0.078	0.035	—						
6. Healthy Diet	0.029	−0.043	−0.068	0.052	−0.162 **	—					
7. Exercise	−0.001	0.012	−0.007	0.043	0.162 **	0.168 **	—				
8. Alcohol Consumption	−0.023	0.073	0.194 **	−0.084	0.156 **	0.000	0.084 ^†^	—			
9. Smoking	−0.135 **	0.011	0.108 *	0.068	0.083	−0.141 **	0.016	0.166 **	—		
10. Education	−0.015	−0.217 **	−0.047	−0.032	−0.071	0.185 **	0.133 **	0.130*	−0.184 **		
11. SES Risk	0.034	0.032	−0.039	−0.039	−0.050	−0.022	−0.010	−0.090	0.135 **	−0.307 **	—
Mean	0.000	0.000	0.000	0.000	0.370	3.306	2.599	0.935	0.743	13.075	0.318
SD	2.972	1.000	2.435	1.000	0.484	1.240	1.159	1.249	1.480	1.724	0.205

^†^*p* < 0.1, * *p* < 0.05, ** *p* < 0.01 (two-tailed tests). Note: Psy Maladjustment = Psychological Maladjustment (age 22–29); SES Risk = Socioeconomic Status Risk (age 10–18); SD = Standard Deviation.

**Table 2 ijerph-19-12034-t002:** Correlation among School Disengagement, Depression, Poor Self-regulatory, Low Self Esteem, Supportive Parenting, and Accelerated Aging (*n* = 386).

	1	2	3	4	5	6
1. Accelerated Aging (age 29)	—					
2. School Disengagement (age 10–18)	0.101 *	—				
3. Depression (age 22–29)	0.139 **	0.122 *	—			
4. Lack of Self-Regulation (age 22–29)	0.168 **	0.238 **	0.439 **	—		
5. Low Self-Esteem (age 22–29)	0.144 **	0.235 **	0.534 **	0.492 **	—	
6. Supportive Parenting (age 10–18)	−0.133 **	−0.337 **	−0.131 *	−0.186 **	−0.300 **	—
Mean	0.000	0.000	2.102	16.047	8.975	0.000
SD	2.972	1.000	1.781	2.927	2.841	1.000

** *p* ≤ 0.01; * *p* ≤ 0.05 (two-tailed tests).

## Data Availability

The data have not been public; however, the data presented in this study are available to request from the corresponding author.

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
