# Peer review of "School Disengagement Predicts Accelerated Aging among Black American Youth: Mediation by Psychological Maladjustment and Moderation by Supportive Parenting"

_ijerph, 2022, doi:10.3390/ijerph191912034_

Round 1

Reviewer 1 Report

The paper presents an interesting article that tries to longitudinally study a relationship between school disengament and a deterioration of people's wellness during the life span. The idea is promising. however, some issues that could improve the quality of research and an easier and more complete understanding by readers should be discussed in greater depth.

- the study lacks contextual elements, for example it should be explained why a sample of black people only was identified, an effect of physical discomfort could also be the effect of continuous discrimination or local segregation, in this case of minority stress. At this point, this hypothesis should also be discussed and contemplated both in the introduction and in the discussion of the results, and studies in this regard should also be cited.

-Indicate the time lapse between measurements between the first and second data collection, enter other information concerning the sample.

- information on research ethics is lacking

Minor revisions:

-It would be advisable to insert an ad hoc paragraph for the policy implications

-The conclusions are too hasty, please argue with final consideration and showing how this study adds to the state of the art.

-

Author Response

1) the study lacks contextual elements, for example it should be explained why a sample of black people only was identified, an effect of physical discomfort could also be the effect of continuous discrimination or local segregation, in this case of minority stress. At this point, this hypothesis should also be discussed and contemplated both in the introduction and in the discussion of the results, and studies in this regard should also be cited

ANS: Thank you for your suggestion. We added paragraphs to better explain contextual elements on pages 1 – 2.

2) Indicate the time lapse between measurements between the first and second data collection, enter other information concerning the sample.

ANS: Thank you for your suggestions. We added information about the time lapse between the first and second waves of data collection and other information concerning the sample on pages 4-5.

 The FACHS sample consisted of 889 Black American families at wave 1 in 1997-1998. All families were in 5th grade at the time of recruitment. At the first wave, 422 families in the sample resided in Georgia and 467 resided in Iowa. All children were in the fifth grade and averaged 10 (SD=.63) years of age. The majority of primary caregivers were females. Approximately 19% had less than a 12th grade education. Almost 65% lived in large urban areas, 15% lived on the suburbs, and 21% lived in rural areas. There were 779 interviews at wave 2 in 1999-2000, 767 at wave 3 in 2001-2002, 714 at wave 4 in 2004-2005, 689 at wave 5 in 2007-2008, and 669 at wave 6 in 2010-2011. Data collection was completed that included blood draws at wave 7 in 2015-2016. Given the logistics of scheduling home visits by phlebotomists, only members of the sample residing in Georgia, Iowa, or a contiguous state were identified as eligible. Thus, after excluding participants who were deceased, incarcerated, or unreachable for some reason, we were left with a pool of 545 individuals, and 470 (86%) participants agreed to be interviewed and provided blood. After excluding samples with poor quality (n=81) and no amplification (n=3), successful assays for mRNA expression were obtained for 386 participants (Mean age = 28.68; females = 242 and males = 144) and were not missing data on the variables of interest. The average annual income of young adults was $22992.75 (SD=17346.44).

3) information on research ethics is lacking.

ANS: Thank you for pointing out this oversight.  We have added a statement at the end of the document stating “Research and Publication Ethics:  All participants gave their informed consent for inclusion before they participated in the study. The study was conducted in accordance with the Declaration of Helsinki, and the protocol was approved by the University of Georgia IRB (Project identification title and number: FACHS Umbrella; 00001153) and also by the University of Iowa IRB (Project identification number 201901770)."

Minor points

4) It would be advisable to insert an ad hoc paragraph for the policy implications.

ANS: Thank you for this suggestion. We have inserted a paragraph for the policy implications on page 11.

 Implications for School Health Policy, Practice, and Equity

Despite its limitations and the need for future studies, the current study provides a better understanding of the link between school disengagement and accelerated aging in adulthood for Black Americans. Our findings imply that school disengagement is a critical adverse experience associated with later accelerated aging and psychological maladjustment. Our findings also suggest that supportive parenting plays a vital role in buffering against the potential threat of painful and lasting consequences of school disengagement including later psychological dysfunction.

This finding is consistent with prior research indicating that Black American parents play an important role in supporting their children in school contexts that may devalue their academic progress (Cole-Lewis et al., 2021; Rowley, Helaire, & Banerjee, 2010), and so provides support for the development of programs to reduce school disengagement as well as support for the use of parenting interventions in the school setting to mitigate long-term outcomes resulting from school disengagement.  These may have the potential to result in better health outcomes for Black Americans in young adulthood and beyond.

5) The conclusions are too hasty, please argue with final consideration and showing how this study adds to the state of the art.

ANS: The current results provide evidence that early experiences of school disengagement may serve as a warning sign for later young adult adjustment difficulties and eventually accelerated aging among African American youth. This problematic trajectory may, in turn, set the stage for health problems later in life. Therefore, our findings provide evidence supporting increased surveillance of these difficulties and increased efforts to utilize this information to support ways to better address school disengagement and better prevent future psychological difficulties and accelerated aging among Black American youth. Future research should consider additional biopsychosocial mechanisms and health behaviors that may also play a role in connecting school disengagement with later mental and physical health problems.  Likewise, future research should consider additional potential moderators including both community and peer influences.

Reviewer 2 Report

1. Congratulations to the presentation of your scientific work. The hypotheses are stated very clearly, and the final results are presented very clearly as well. You find a view comments below.

2. In chapter "2.1 Sample" you say that you only use data with no missing variables of interest. It would be interesting to get a short description of the sample of data not used for the analyses concerning the confounders and the total number of excluded cases. The question whether certain groups did not answer the variables of interest can be important for the final interpretation.

3. Line 161: I suppose, you mean 14.36 years of age?  The number 14.36 is not clear

4. In chapter "2.2 Methods" it would be good to clarify whether you standardized ""school disengagement" and "supportive parting", which are presented as sum scores at different waves, to a total measure with mean=0 and SD=1. It seems to be so in table 1 and table 2. What is the reason for this standardizing process and how did you proceed exactly?

5. All variables are presented in terms of mean and SD. Confidence intervals seem to be symmetric. Did you check in any way for normal distribution?

Author Response

1) In chapter "2.1 Sample" you say that you only use data with no missing variables of interest. It would be interesting to get a short description of the sample of data not used for the analyses concerning the confounders and the total number of excluded cases. The question whether certain groups did not answer the variables of interest can be important for the final interpretation.

ANS: Thank you for this suggestion. We now better describe patterns of attrition in “Section 2.1. Sample” on page 4 and “2.3 Analytic Strategy” section on pages 6-7

 At the first wave, the sample resided in Georgia was 422, and Iowa was 467. All children were in the fifth grade and averaged 10 (SD=.63) years of age. The majority of primary caregivers were females. Approximately 19% had less than a 12th-grade education. Almost 65% lived in large urban areas, 15% lived in the suburbs, and 21% lived in rural areas. There were 779 interviews at wave 2 in 1999-2000, 767 at wave 3 in 2001-2002, 714 at wave 4 in 2004-2005, 689 at wave 5 in 2007-2008, and 669 at wave 6 in 2010-2011. Data collection was completed that included blood draws at wave 7 in 2015-2016. Given the logistics of scheduling home visits by phlebotomists, only members of the sample residing in Georgia, Iowa, or a contiguous state were identified as eligible. Thus, after excluding participants who were deceased, incarcerated, or unreachable for some reason, we were left with a pool of 545 individuals, and 470 (86%) participants agreed to be interviewed and provided blood. After excluding samples with poor quality (n=81) and no amplification (n=3), successful assays for mRNA expression were obtained for 386 participants (Mean age = 28.68; females = 242 and males = 144) and were not missing data on the variables of interest.  

 We compared those retained for our analyses (N = 386) with those who were lost to attrition (N = 503) on all study variables used at baseline.  We found that only gender was significantly different for those lost to attrition, with more males than females lost to attrition.

2) Line 161: I suppose, you mean 14.36 years of age?  The number 14.36 is not clear

ANS: Thank you for catching this. We made the correction “When the participants were the mean age of 10.41, 12.29, 15.67, and 18.77 at Waves 1 – 4” on page 4.

3) In chapter "2.2 Methods" it would be good to clarify whether you standardized ""school disengagement" and "supportive parting", which are presented as sum scores at different waves, to a total measure with mean=0 and SD=1. It seems to be so in table 1 and table 2. What is the reason for this standardizing process and how did you proceed exactly?

ANS:

After averaging items across items to form measures of school disengagement and supportive parenting, the measures were standardized (i.e., a mean of 0 and a standard deviation of 1).  The school disengagement and supportive parenting measures were standardized to allow the calculation of interaction terms that would not be correlated with the main effects, reducing the potential for multicollinearity among variables in the model (Lei & Simons, 2021).  In addition, standardized scores make it easier to read the results (Hunter & Hamilton, 2002), easier to interpret the simple slopes, and easier to test slopes (Dawson & Richter, 2006).  We have now expanded our description of the scales in section 2.2.

 4) All variables are presented in terms of mean and SD. Confidence intervals seem to be symmetric. Did you check in any way for normal distribution?

ANS: Thank you for asking. Yes. We checked the distribution of our accelerated aging measure using the Kolmogorov-Smirnov and Shapiro Wilk tests as follows: 

Tests of Normality

Kolmogorov-Smirnov

Shapiro Wilk

Statistic

df

Sig.

Statistic

df

Sig.

Accelerated Aging

.033

386

.200

.997

386

.739

Reviewer 3 Report

ABSTRACT

I suggest to report participants’ mean age and the percentages of males and females.

INTRODUCTION    

I really appreciate the clear and detailed way by which authors argue the impact of school disengagement on negative outcomes; however I suggest to authors to describe in the first introductory paragraph the study’s research questions and its overall aim.

1.1.Accelerated Aging

I suggest to authors to change the name of tis paragraph in “accelerated aging measure” considering the fact that they describe a measure used in the study, Furthermore, I also suggest to describe in more detailed way the phenomenon of the accelerated aging.

1.2.Supportive Parenting as a Moderator

I suggest to authors to better define the construct of parenting and parenting styles. Furthermore, in this framework, I also suggest to better describe the impact of parental school involvement on youths’ (mal)adjustment.

I also suggest to add a paragraph to describe the Black American culture.

1.3.Current Study and Hypotheses

I appreciate the clear way by which authors describe the study’s hypotheses. However, I suggest to better anchor them to the study’s objectives and literature.

2.1. Sample

Please, describe the socio-demographic characteristics of the sample (e.g. income, parents’ education).

2.3. Analytic strategy

Please describe the examined conceptual model (predictors, outcomes, covariates, mediator, moderator, covariates).

3.1. Descriptive and Association Analysis

Please report also the direction of the significant associations.

Results, discussions and study implications are very well written.

Author Response

1) ABSTRACT:  I suggest to report participants’ mean age and the percentages of males and females.

ANS: Thank you for the suggestion. We have added the participants’ mean age and the percentages of males and females in the Abstract.

2) INTRODUCTION : I really appreciate the clear and detailed way by which authors argue the impact of school disengagement on negative outcomes; however I suggest to authors to describe in the first introductory paragraph the study’s research questions and its overall aim.

ANS: Thank you for the suggestion. “The purpose of the current study is to investigate the impact of school disengagement on accelerated aging among Black Americans and examine whether it is mediated through psychological maladjustment and buffering by supporting parenting” had been added on page 2.

3) 1.1.Accelerated Aging : I suggest to authors to change the name of this paragraph in “accelerated aging measure” considering the fact that they describe a measure used in the study, Furthermore, I also suggest to describe in more detailed way the phenomenon of the accelerated aging.

ANS: Thanks for recommending it. 1. We have changed the name to Accelerated Aging Measure. 2. We also added additional detail on pages 2- 3:

Peters et al. [27] found strong agreement between the discovery sample of 7074 individuals of European ancestry and a replication sample of 7909 that included various ethnic groups including African Americans, Native Americans, and Hispanics. Corresponding to earlier epigenetic clock measures of Horvath [28]and Hannum et al. [29], the transcriptome index is highly correlated with age and the difference between an individual’s predicted and chronological age indicates, in years, the extent to which they are experiencing accelerated or decelerated aging.

Moreover, it is related to metabolic functioning, immune senescence, DNA repair, inflammation, and mitochondrial effectiveness, implicating the development of cardiometabolic problems and chronic illnesses, such as diabetes, coronary heart disease, stroke, and Alzheimer’s [27]. This evidence indicates that accelerated aging, assessed by the transcriptome index, consists of many gene expression changes that are associated with the onset of chronic illness [31].

 4) 1.2.Supportive Parenting as a Moderator: I suggest to authors to better define the construct of parenting and parenting styles. Furthermore, in this framework, I also suggest to better describe the impact of parental school involvement on youths’ (mal)adjustment.

ANS: Thank you for your suggestion. We have better defined our constructs and the way they impact youth maladjustment on pages 3-4:

Parenting styles, including authoritarian, authoritative, and permissive domains [47], have been shown to relate to teenagers’ psychological problems, behavioral problems, and academic performance [e.g., 47, 48]. Authoritative parenting emphasizes warmth, involvement, responsiveness, and encouraging autonomy. Permissive parenting makes few demands, exhibits noncontrolling behaviors, and uses minimal punishment. Authoritarian parenting characterizes a high level of directive, restriction and rejection, and power-asserting behaviors [47].

Likewise, supportive parenting has been linked to improved psychological factors in adolescence [7, 37, 46]. For instance, Frazer and Fite [19] found that parents’ involvement in their children’s lives was negatively related to depression symptoms and may buffer the impact of adverse experiences on depression. Supportive parenting is positively related to self-esteem [49] and cognitive and emotional self-control [7]. For example, when parents and teenagers engage in supporting interactions, teenagers learn parents' cognitive control and problem-solving skills through observation and modeling [7, 50].

5) 1.3.Current Study and Hypotheses: I appreciate the clear way by which authors describe the study’s hypotheses. However, I suggest to better anchor them to the study’s objectives and literature.

ANS: Thank you for this suggestion.  We have added a brief section to better anchor the hypotheses.  Just before the specific hypotheses we now say “Based on the preceding literature we formulated three specific hypotheses reflecting our expectation of longitudinal associations between school disengagement and problems manifesting in young adulthood and the potential protective role of parenting”

6) Sample: Please, describe the socio-demographic characteristics of the sample (e.g. income, parents’ education).

ANS: Thank you for your suggestions. We described the socio-demographic characteristics of the sample (e.g. income, parents’ education) in “Section 2.1. Sample” on page 4.

At the first wave, the sample resided in Georgia was 422 and Iowa was 467. All children were in the fifth grade and averaged 10 (SD=.63) years of age. The majority of primary caregivers were females. Approximately 19% had less than a 12th grade education. Almost 65% lived in large urban areas, 15% lived on the suburbs, and 21% lived in rural areas. Average annual income of young adults was $22992.75 (SD=17346.44).

7) Analytic strategy: Please describe the examined conceptual model (predictors, outcomes, covariates, mediator, moderator, covariates).

ANS: We revised the Analytic strategy section. We performed path modeling in Mplus 8 [62] to examine the indirect effect of early school disengagement on accelerated aging through young adult psychological maladjustment and to examine the stress-buffering effects by supportive parenting. We first examined the direct effect of school disengagement across Wave 1 through 4 on accelerated aging, averaging scores at Waves 1 to 4 to form school disengagement. Next, using path modeling to assess mediation effect, Mplus was used to generate bootstrapped confidence interval (CI). The significance of hypothesized indirect effects was assessed by examining the 95% confidence interval (CI) estimated with bias-corrected and accelerated bootstrapping with 1,000 replications. To evaluate the goodness-of-fit of the model, we report the standardized root mean square residual (SRMR < .05) and the comparative fit index (CFI > .90) as well as the Chi-square test and degrees of freedom. Third, we tested the moderated mediation hypotheses using path modeling. We examined the interaction of school disengagement and supportive parenting in predicting psychological maladjustment. After averaging items across items to form measures of school disengagement and supportive parenting, the measures were standardized (i.e., a mean of 0 and a standard deviation of 1).  The school disengagement and supportive parenting measures were standardized to allow the calculation of interaction terms that would not be correlated with the main effects, reducing the potential for multicollinearity among variables in the model (Lei & Simons, 2021).  In addition, standardized scores make it easier to read the results (Hunter & Hamilton, 2002), easier to interpret the simple slopes, and easier to test slopes (Dawson & Richter, 2006).To explicate the shape of a significant interaction effect, we plotted the simple slopes with school disengagement and psychological maladjustment, showing the effect of school disengagement on psychological maladjustment with confidence intervals around the regression lines for high vs. low supportive parenting and reporting conditional indirect effects. Finally, we investigated gender differences using the multiple group analysis option in Mplus. We included gender, education, healthy diet, exercise, alcohol consumption, and smoking as covariate variables in all analyses.

 8) Descriptive and Association Analysis:  Please report also the direction of the significant associations.

ANS: Thank you for your recommendation. We added the direction of the significant associations. The mean, standard deviation, and zero-order correlations for the study variables are shown in Table 1. As expected, school disengagement in childhood was positively correlated with young adult psychological maladjustment (r = .244, p = < .001) and accelerated aging (r = .101, p = .048), negatively correlated with supportive parenting (r = -.337, p = <.001) and education (r = -.217, p = <.001). In addition, accelerated aging was positively related to psychological maladjustment (r = .185, p = <.001), negatively related to supportive parenting (r = -.133, p = .009), and smoking (r = -.135, p = .008). Moreover, psychological maladjustment was negatively related to supportive parenting (r = -.253, p = <.001), positively related to alcohol consumption (r = .194, p = <.001), and smoking (r = .108, p = .033). School disengagement and accelerated aging were positively associated with depression, poor self-regulation, and low self-esteem, and supportive parenting was negatively correlated to them, as shown in Table 2.

Round 2

Reviewer 2 Report

The Requests of the Revision have been answered clearly. Thank you very much